# The effects of sport-specific training on individuals action strategies while avoiding a virtual player approaching on a 45° angle while completing a secondary task

Brooke J. Thompson, Michael E. Cinelli[*]

Department of Kinesiology & Physical Education, Wilfrid Laurier University, Waterloo, ON, Canada

These authors contributed equally to this work.
* mcinelli@wlu.ca

**Data Availability Statement:** The raw data is uploaded to Open Science Forum (https://osf.io/dx27e/).

## Abstract

Sports provide varying scenarios where athletes must interact with and avoid opposing players in dynamic environments. As such, sport-specific training can improve one's ability to integrate visual information which may result in improved collision avoidance behaviours. However, improved visuomotor capabilities are highly task dependent (i.e., athletes must be tested in sport-specific settings). The current study examined whether sport-specific training influenced individuals' collision avoidance behaviours during a sport-specific task in virtual reality. Untrained young adults (N = 21, 22.9±1.9 yrs, 11 males) and specifically trained athletes (N = 18, 20±1.5 yrs, 7 males) were immersed in a virtual environment and were instructed to walk along a 7.5m path towards a goal located along the midline. Two virtual players positioned 2.83m to the left and right of the midline approached participants on a 45° angle at one of three speeds: 0.8x, 1.0x, or 1.2x each participant's average walking speed. Participants were instructed to walk to a goal without colliding with the virtual players while performing a secondary task; reporting whether a shape changed above either of the virtual players' heads. Results revealed that athletes had a higher percentage of correct responses on the secondary task compared to untrained young adults. However, there was no group differences in the average time to first avoidance or average minimum clearance, but athletes were more variable in their avoidance behaviours. Findings from this study demonstrate that athletes may be more adaptive in their behaviours and may perform better on attentionally demanding tasks in dynamic environments.

## Introduction

Avoiding collisions with other humans is a necessary component of everyday life and is especially important in many sport settings. Athletes possess improved visual strategies which allow them to effectively identify and integrate environmental information and use it to produce appropriate actions [1]. During locomotion, the visual system provides valuable

**Funding:** This study was funded by a grant award to MEC from the Natural Sciences and Engineering Research Council of Canada (NSERC-2019-05894). The funders are a national organization and had no role in study design, data collection and analysis, decision to publish, or preparation of the manuscript.

information from a distance to allow individuals to make anticipatory adaptive behaviours (i.e., alter trajectory and/or gait speed) to successfully avoid oncoming collisions [2,3]. To initiate an adaptive behaviour change at the appropriate time, humans rely on optical variables such as time-to-contact (TTC), which is estimated from the inverse rate of dilation of an object's image on the retina (i.e., tau) [4]. As the retinal image of an approaching object increases, individuals may alter their gait speed and/or trajectory once the image size reaches an optical expansion threshold [5]. If individuals rely on a consistent threshold to determine their timing of avoidance, then the TTC between them and an approaching object will also be constant [6,7]. For instance, Pfaff and Cinelli [7] showed that young adults maintained a consistent TTC when avoiding a pedestrian approaching along an unpredictable path, which suggests that they were using a consistent optical expansion threshold to determine when to initiate an avoidance behaviour.

After determining *when* to initiate an avoidance behaviour, *how* one controls their actions to avoid a collision is critical to their success. During obstacle avoidance, it is thought that humans maintain a protective zone (i.e., personal space) by controlling the distance between them and an approaching obstacle or person [8]. The protective zone has been found to be a consistent size regardless of one's walking speed and allows for time to perceive, evaluate, and react to potential perturbations within the environment [5,8]. The protective zone is thought to be elliptical in shape [8], suggesting that the angle in which an object approaches affects individuals' avoidance behaviours. More specifically, previous research has found that avoiding a stationary or moving obstacle on a 180˚ collision course requires a simple path adjustments [5,7,9]. Conversely, avoiding obstacles or pedestrians approaching on acute angles (45˚ or 90˚) angle involves the coordination of more complex collision avoidance strategies involving both path and speed adjustments [10–12].

Sports provide varying scenarios where athletes must interact with and avoid opposing players approaching at acute angles ($< 90˚$) while simultaneously completing multiple tasks. As a result of their sport experiences, specifically trained athletes have adapted their collision avoidance behaviours resulting in an improved ability to extract important information from the environment [1]. Pfaff and Cinelli [6] compared the action strategies of specifically trained rugby players and non-athletes while they avoided colliding with an approaching pedestrian who walked along an unpredictable pathway. It was found that the specifically trained athletes both consistently and accurately avoided collisions later compared to their non-athlete counterparts [6]. The authors suggested that sport-specific training may adjust one's optical expansion threshold based on knowledge of their own capabilities, which allows athletes to successfully avoid collisions later [5,6]. Since avoiding an opponent later, protects one's movement decisions, improved perception-action skills provide athletes with a competitive advantage in sport-specific contexts [6]. In addition to improved visuomotor integration skills, athletes have enhanced perceptual-cognitive skills, such as effective working memory and attention allocation, compared to untrained individuals [13]. As such, athletes tend to be more successful in performing simultaneous tasks as a result of their sport-specific training [14–16].

Although it is evident that athletes possess improved perceptual-cognitive and perceptual-motor capabilities, evidence suggests that these improved capabilities are highly task-dependent and is related to the specificity of their training [6,17–19]. For instance, previous research has demonstrated that athletes perform better when they are tested in environments that are similar to their sport-specific training [17]. The current study aims to build upon the previous collision avoidance literature by examining how sport-specific training influences individuals' action strategies during a sport-specific task in virtual reality. To make the task closer to a sport-specific scenario, the participants were asked to avoid a virtual player approaching on a 45˚ angle while completing an attentionally demanding secondary task. As a result of their

sport-specific training experience, it was expected that athletes would consistently avoid collisions later while making fewer errors on the secondary task compared to the controls.

## Methodology

### Participants

Eighteen varsity athletes (11 Females, age: 20 ± 1.53 years) and twenty-one untrained young adults (10 Females; age: 22.9 ± 1.88 years) participated in the current study (**Table 1**). Participants were recruited from November 1st, 2022, to April 3rd, 2023. The athlete group included varsity athletes on the current roster for soccer, basketball, rugby, football, or hockey. The untrained young adults included individuals who had not trained for a sports team at a representative or varsity level after high school. Participants in both groups were excluded if any of the following were present: (a) cognitive, neurological, or sensory impairments influencing everyday activities; (b) musculoskeletal injuries or disorders that limited their ability to stand or walk comfortably for 1 hour; (c) abnormal or uncorrected vision; (d) a concussion in the last 2 years and/or (e) absence of stereoscopic vision. This experiment was reviewed and approved by the university's Research Ethics Board. Informed written consent was obtained by all participants on Qualtrics, a university-approved online survey platform, prior to participation.

### Experimental set-up

The experiment was conducted in a large rectangular room with a 7.5-meter (m) pathway cleared along the midline. To begin, participants were outfitted with an HTC VivePro 2 head mounted display (HMD) (wireless attachment allowed free movement), which provided an immersive virtual environment (VE) where participants performed a collision avoidance task. In the VE, a visible goal (white square) was located at the end of the 7.5m pathway directly in line with the participants' home target (starting location). The HTC VivePro 2 HMD collected the participants' positional data at a sampling frequency of 90 Hz. The HMD is an effective method of recording the X, Y, and Z coordinates of the virtual player (VP) and the participant to monitor their positions relative to each other throughout the experiment, which was necessary for the calculation of dependent variables related to the participants' avoidance behaviours.

### Protocol

Participants began by completing four baseline walking trials where they were asked to "walk at a comfortable pace from their home target to the goal", which was used to calculate the average walking speed for each participant. The remainder of the trials were separated into two blocks: 1) familiarization block; and 2) experimental block, where participants performed a collision avoidance task. For the two blocks of trials, VPs were located 7.5m away from the participants' home target and 2.83m to the left and right of the midline (LVP and RVP starting positions respectively) (**Figs 1 & 2**). During the familiarization block, one VP appeared at either starting position (left or right). The participants were instructed to begin walking towards the goal. Once the participant walked 1m, one VP (LVP or RVP) began approaching the participants at 45˚ angle towards the intersection point located 5m from the participant's home target at one of three speeds, 0.8x (slow), 1.0x (normal), or 1.2x (fast) each participant's average walking speed. Varying speeds were utilized to examine whether participants were relying on a consistent optical expansion threshold to control their timing of avoidance. The participants were instructed to "walk to the goal while avoiding colliding with the VP." Block 1 consisted of six randomized trials (1 trial x 3 VP speeds x 2 VP starting locations).

**Table 1. Participant characteristics.**

| Participant | Sex (Male/Female) | Age (Years) | Height (Centimeters) | Weight (Kilograms) | Sport |
|---|---|---|---|---|---|
| 1 | Female | 22 | 180 | 63.5 | --- |
| 2 | Female | 24 | 163 | 56.7 | --- |
| 3 | Female | 20 | 160 | 65.8 | --- |
| 4 | Female | 22 | 160 | 47.6 | --- |
| 5 | Female | 21 | 170 | 66.0 | --- |
| 6 | Female | 26 | 163 | 59.0 | --- |
| 7 | Female | 23 | 170 | 58.0 | --- |
| 8 | Female | 22 | 168 | 62.0 | --- |
| 9 | Female | 22 | 163 | 61.2 | --- |
| 10 | Female | 24 | 168 | 90.7 | --- |
| 11 | Male | 26 | 174 | 87.0 | --- |
| 12 | Male | 22 | 175 | 80.0 | --- |
| 13 | Male | 22 | 185 | 81.7 | --- |
| 14 | Male | 21 | 173 | 74.8 | --- |
| 15 | Male | 20 | 179 | 66.0 | --- |
| 16 | Male | 23 | 185 | 95.0 | --- |
| 17 | Male | 23 | 167 | 78.5 | --- |
| 18 | Male | 26 | 172 | 65.7 | --- |
| 19 | Male | 26 | 178 | 86.2 | --- |
| 20 | Male | 24 | 180 | 97.0 | --- |
| 21 | Male | 23 | 183 | 74.8 | --- |
| Average | ---------- | 22.9 | 172.0 | 72.2 | ---------- |
| SD | ---------- | 1.88 | 7.89 | 13.7 | ---------- |
| 1 | Female | 20 | 172.7 | 61.2 | Hockey |
| 2 | Female | 18 | 167.6 | 61.2 | Hockey |
| 3 | Female | 18 | 167.6 | 70.3 | Hockey |
| 4 | Female | 19 | 170.2 | 61.2 | Hockey |
| 5 | Female | 21 | 165.1 | 61.7 | Hockey |
| 6 | Female | 20 | 157.5 | 57.2 | Hockey |
| 7 | Female | 21 | 167.6 | 54.4 | Soccer |
| 8 | Female | 19 | 174.0 | 79.4 | Hockey |
| 9 | Female | 18 | 162.5 | 61.2 | Hockey |
| 10 | Female | 18 | 162.5 | 56.7 | Hockey |
| 11 | Female | 19 | 165.1 | 54.4 | Hockey |
| 12 | Male | 22 | 185.4 | 95.3 | Rugby |
| 13 | Male | 23 | 183.0 | 83.0 | Football |
| 14 | Male | 21 | 200.0 | 95.3 | Basketball |
| 15 | Male | 20 | 177.0 | 71.7 | Soccer |
| 16 | Male | 21 | 190.5 | 80.7 | Soccer |
| 17 | Male | 20 | 187.0 | 86.2 | Soccer |
| 18 | Male | 22 | 196.6 | 113.4 | Football |
| Average | ---------- | 20.0 | 175.1 | 72.4 | ---------- |
| SD | ---------- | 1.53 | 12.5 | 17.0 | ---------- |

Block 2 (experimental block) was the same as Block 1, however, LVP and RVP appeared at their starting positions with only one approaching the participant within a given trial. In addition, to make the task similar to sporting scenarios, a secondary task was added to the

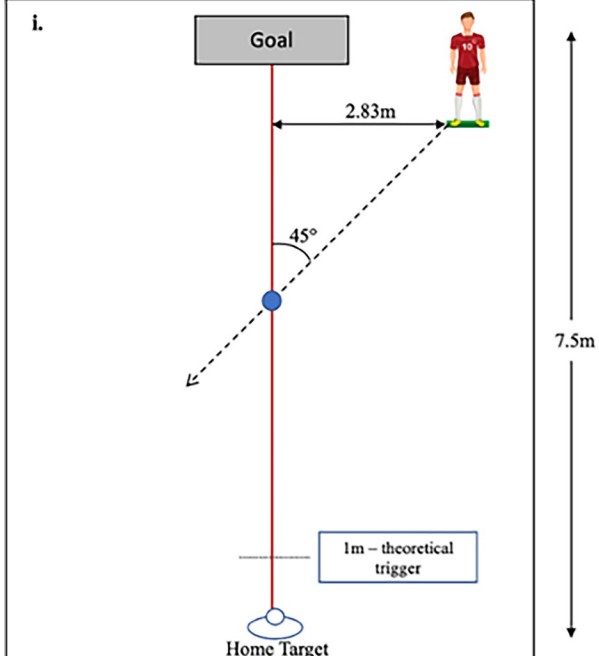 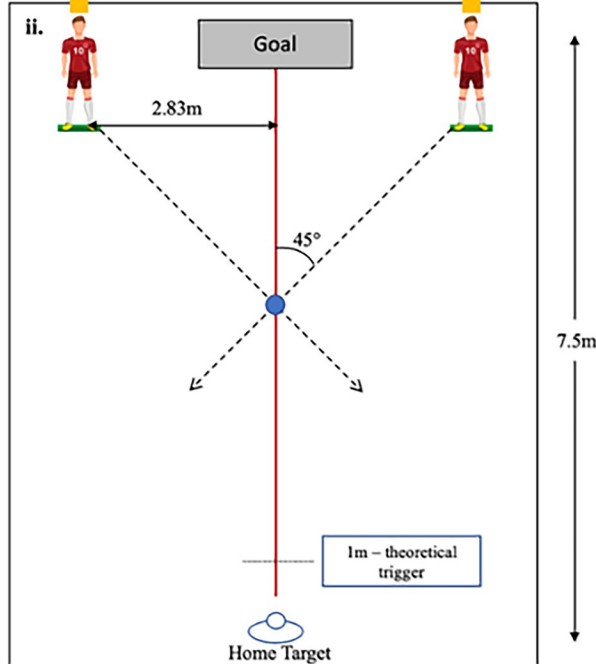

**Fig 1. Experimental set-up for the (i) familiarization trials and (ii) experimental trials.** The distance between the home target and the goal was 7.5m. In the familiarization trials, one virtual player was located 2.83m to the right or left of the midline and 4m from the intersection point (blue circle). The set-up was similar for the experimental trials, with two virtual players located 2.83m to the left and right of the midline.

experimental trials. On half of the experimental trials, a yellow square located above the stationary VP would change to one of four shape options (diamond, triangle, circle, or hexagon), 2s after the VP began moving (participant had walked 1m) and would remain changed for 0.5s before changing back to a square (creating a secondary task). On the remaining half of trials, the shape would remain unchanged (yellow square) for the duration of the trial. Participants were asked to "walk to their goal while avoiding colliding with the VP" and to "report whether a shape change occurred above either of the VPs' heads" once they arrived at the goal. Instructions as to how or when to avoid the VP were not provided and the participants were asked to successfully perform both the collision avoidance and the secondary task, but were not instructed to prioritize one task over the other. Block 2 included 60 randomized trials (5 trials x 3 VP speeds x 2 VP starting locations x 2 attention tasks) for a total of 70 walking trials (4 baseline + 6 familiarization + 60 experimental) in the experiment.

## Data analysis

The location of each participant's head in space was estimated using the HTC VivePro2 HMD. This estimate allowed for the calculation of two dependent variables: 1) the average time to first avoidance (seconds); and 2) minimum clearance (meters) (**Fig 3**). To examine whether sport-specific training influenced the timing in which individuals initiate an avoidance behaviour, the *average time to first avoidance behaviour* was calculated. This was determined by calculating the time between when the VP began moving to when the participant made the first avoidance behaviour (i.e., change in speed and/or path). A change in walking speed was the point at which the participant's walking speed fell and remained outside of two standard deviations (SD) of their average approach phase walking speed. The average walking speed for the

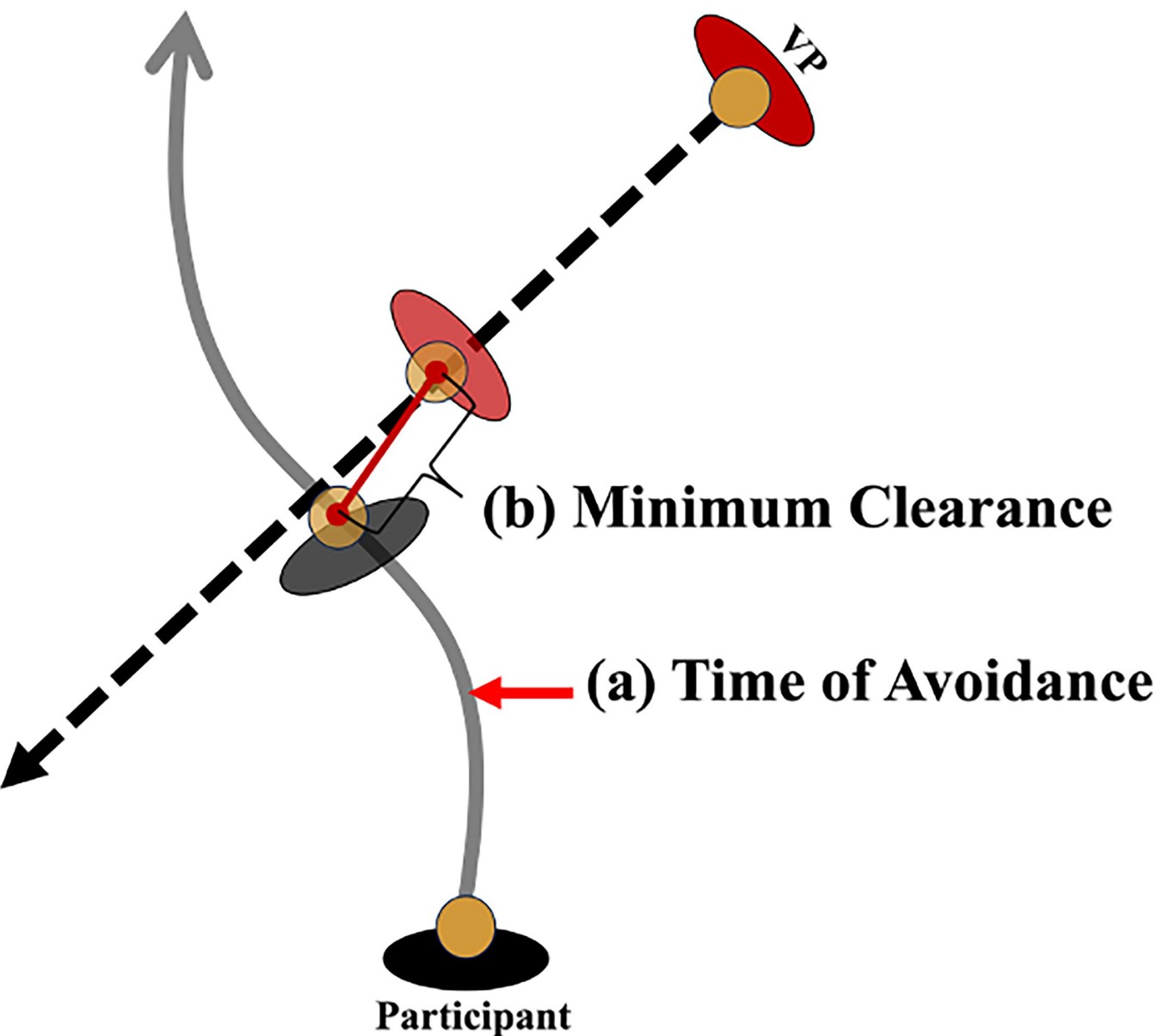

**Fig 2. Virtual environment simulating the inside of a stadium where participants completed the task.**

approach phase was calculated for 1s after the participant exceeded 40cm/s. A change in path was the point at which the participant's medial-lateral (ML) position fell and remained outside of two standard deviations of their average approach phase ML position. The average ML position during the approach phases was calculated for 1s after the participant exceeded 40cm/s. To examine whether sport-specific training influenced one's personal space, *minimum clearance* distance was examined. Minimum clearance was defined as the minimum distance between the participants' and VP's positions in space within a trial. Average minimum clearance and time of first avoidance were calculated for each participant across the 10 trials of each VP approach speed and VP start location (collapsed across attention tasks). In addition, to examine the consistency of the collision avoidance behaviours between groups, the variability (SD) of minimum clearance and time of first avoidance across the 10 trials was calculated.

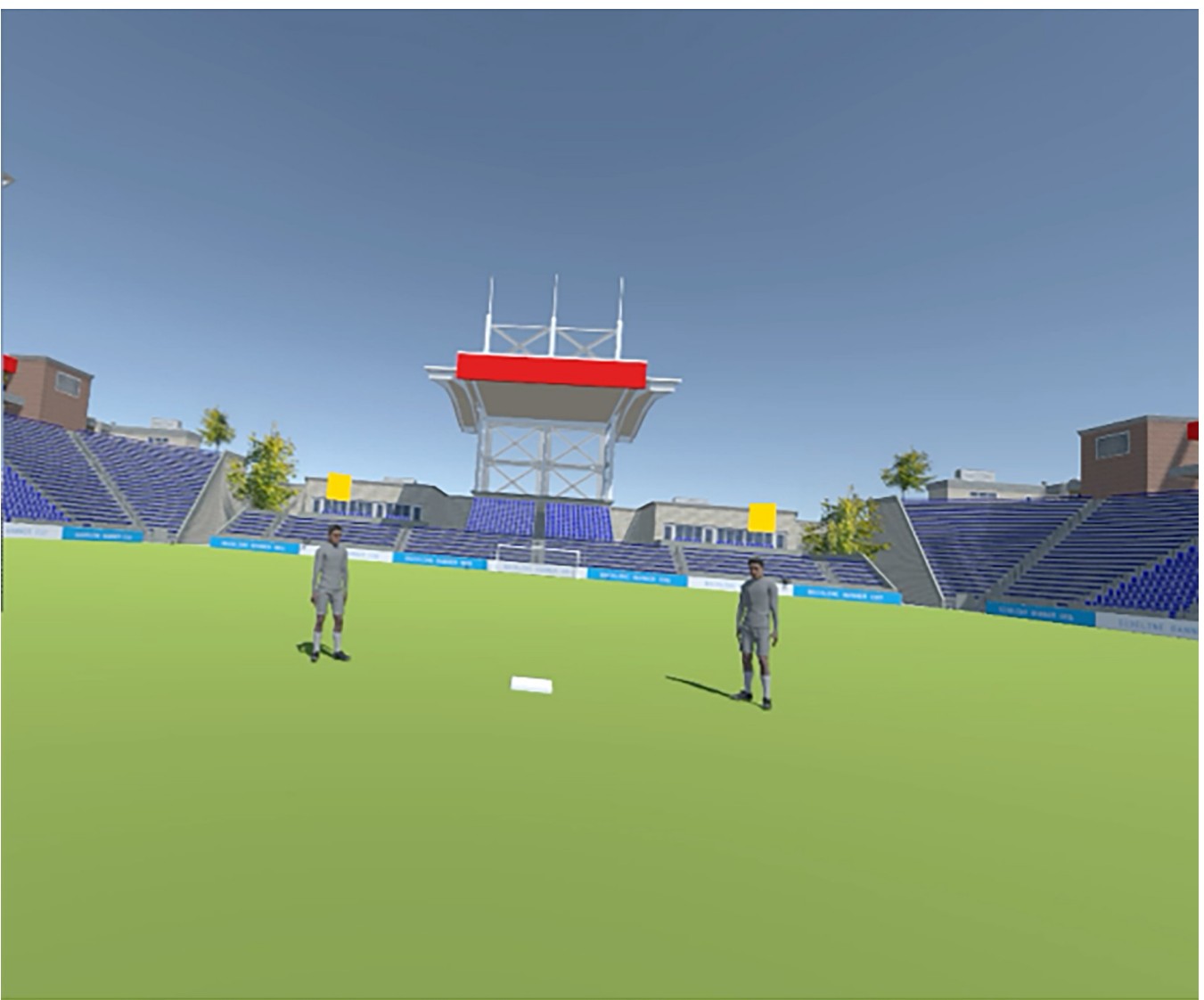

**Fig 3.** Visual representation of the calculations for the outcome variables **(a)** time of first avoidance and **(b)** minimum clearance distance.

Lastly, the percentage of correct responses on the secondary task were calculated for each participant by dividing the number of correct responses by the total number of trials completed.

## Statistical analysis

IBM SPSS Statistical software was used to conduct the statistical analyses. Data were assessed for normality, sphericity, and homogeneity of variances. Where appropriate, the Greenhouse-Geisser correction was applied for any violations of sphericity. Separate three-way mixed repeated measures ANOVAs were conducted to examine whether time to first avoidance (average and variability) and minimum clearance (average and variability) differed between groups when the direction (2 levels: left and right) and speed (3 levels: slow, normal, and fast) of the approaching VP were manipulated. In addition, to examine whether performance on the secondary task (measured as a percent of correct responses) differed between groups, an independent samples t-test was performed. One outlier was removed prior to running the

independent samples t-test. The alpha value was set to p < .05. Effect size for significant effects was reported using partial eta squared ($\eta_p^2$) for the repeated measures ANOVAs (0.01 was small, 0.06 was medium, and 0.14 was large), and Cohen's d (d) for the independent samples t-test (0.2 was small, 0.5 was medium, and 0.8 was large). Data are presented as mean ± standard deviation. Bonferroni corrected pairwise comparisons were examined to identify where the significant differences existed.

## Results

The average walking speed during the approach phase did not significantly differ between athletes (1.06 ± .13 m/s) and controls (1.14 ± .14m/s). It was found that the percentage of total trials in which a speed change proceeded a path change was 65% for the athletes and 72% for the controls.

### Time to first avoidance

Contrary to our hypothesis, there was no significant difference in the time in which an avoidance behaviour was initiated between athletes (1.66 ± .324s) and controls (1.67 ± .343s), F(1, 37) = .02, p = .890, $\eta_p^2$ = .001 (**Fig 4**). Moreover, there were no significant interactions or main effects of direction or speed on average time to first avoidance, suggesting that all participants initiated an avoidance behaviour at a similar time regardless of group or condition (**Fig 5**).

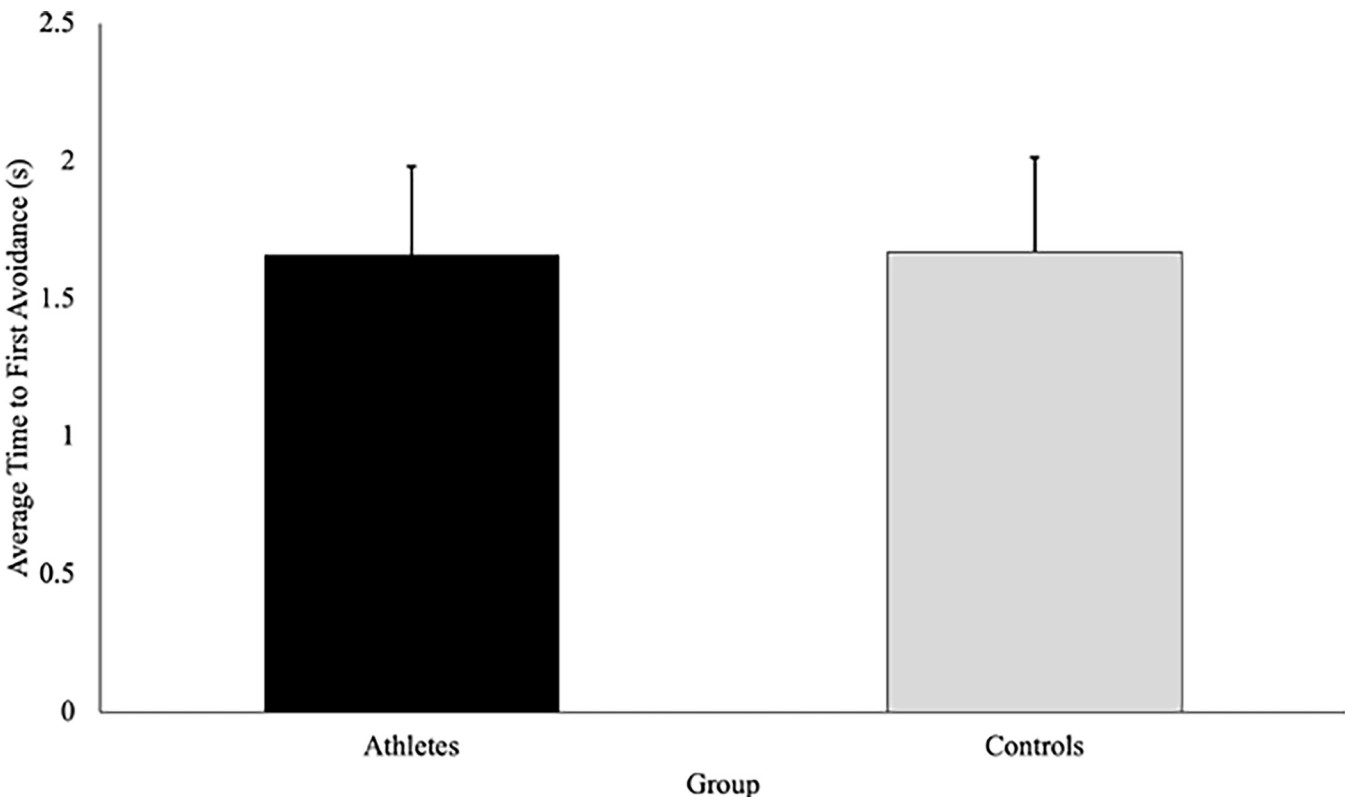

**Fig 4. Time to first avoidance (seconds; with SD bars) is a measure of the elapsed time between when the participants passed the trigger (1m) to when they initiated a behaviour change to avoid the virtual player.** This figure shows that there was no difference in the time in which an avoidance behaviour was initiated between athletes and controls (p = 890).

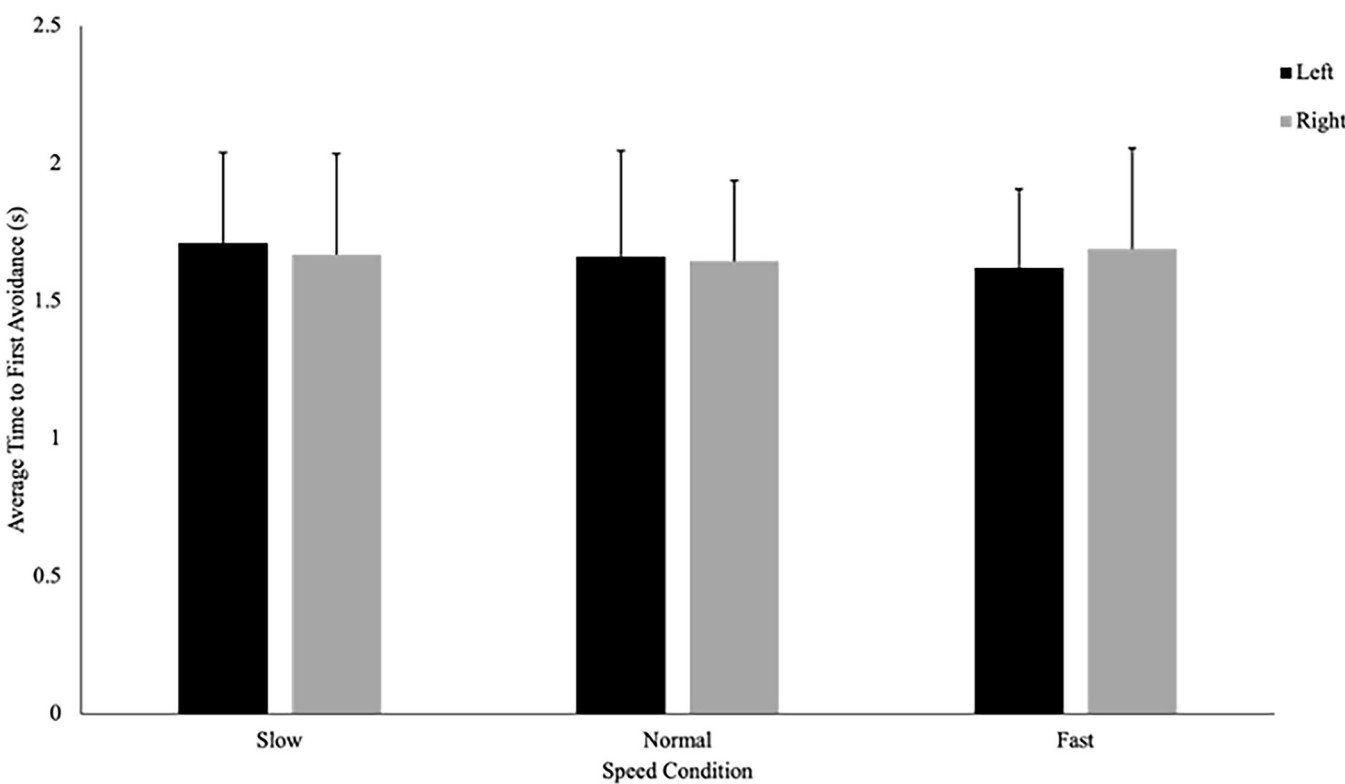

**Fig 5. There was no influence of the direction (left or right) or speed (slow, normal, or fast) of the approaching virtual player on the average time to initiate an avoidance behaviour.**

There was no significant difference in the variability in time to first avoidance between athletes (.487 ± .038s) and controls (.461 ± .035s), $F(1,37)$ = .26, p = .613, $\eta_p^2$ = .007 (**Fig 6**). There was a significant main effect of speed on the variability in time of avoidance, $F(2, 36)$ = 10.32, p < .001, $\eta_p^2$ = .364. Pairwise comparisons revealed that variability was greater when the VP approach at the slow speed (.548 ± .035s), compared to the normal (.462 ± .031s) and fast speeds (.413 ± .023s) (p = .003 and p < .001 respectively). Further, the variability was significantly higher in the normal speed when compared to the fast speed (p = .032). There were no significant interactions or main effects of direction on the variability of the time to first avoidance behaviour (**Fig 7**).

## Minimum clearance

There was no significant difference in the average minimum clearance maintained by athletes (.991 ± .186m) and controls (.934 ± .203m), $F(1,37)$ = 1.05, p = .311, $\eta_p^2$ = .028 (**Fig 8**). There was a statistically significant main effect of speed on average minimum clearance, $F(1.12, 44.36)$ = 14.80, p < .001, $\eta_p^2$ = .286. Pairwise comparisons revealed that average minimum clearance was significantly greater when the VP approached at the slow speed (1.03 ± .161m) compared to the normal (.940 ± .198m) and fast (.921 ± .214m) walking speeds (p < .001, and p = .002 respectively) (**Fig 9**). There were no significant differences between the minimum clearance maintained by the participants in the normal and fast speeds (p = .237). Additionally, no significant interactions or main effects of direction were observed for average minimum clearance.

The athlete group (.130 ± .009m) was significantly more variable in their minimum clearance compared to the controls (.103 ± .008m), $F(1,37)$ = 5.08, p = .030, $\eta_p^2$ = .121 (**Fig 10**).

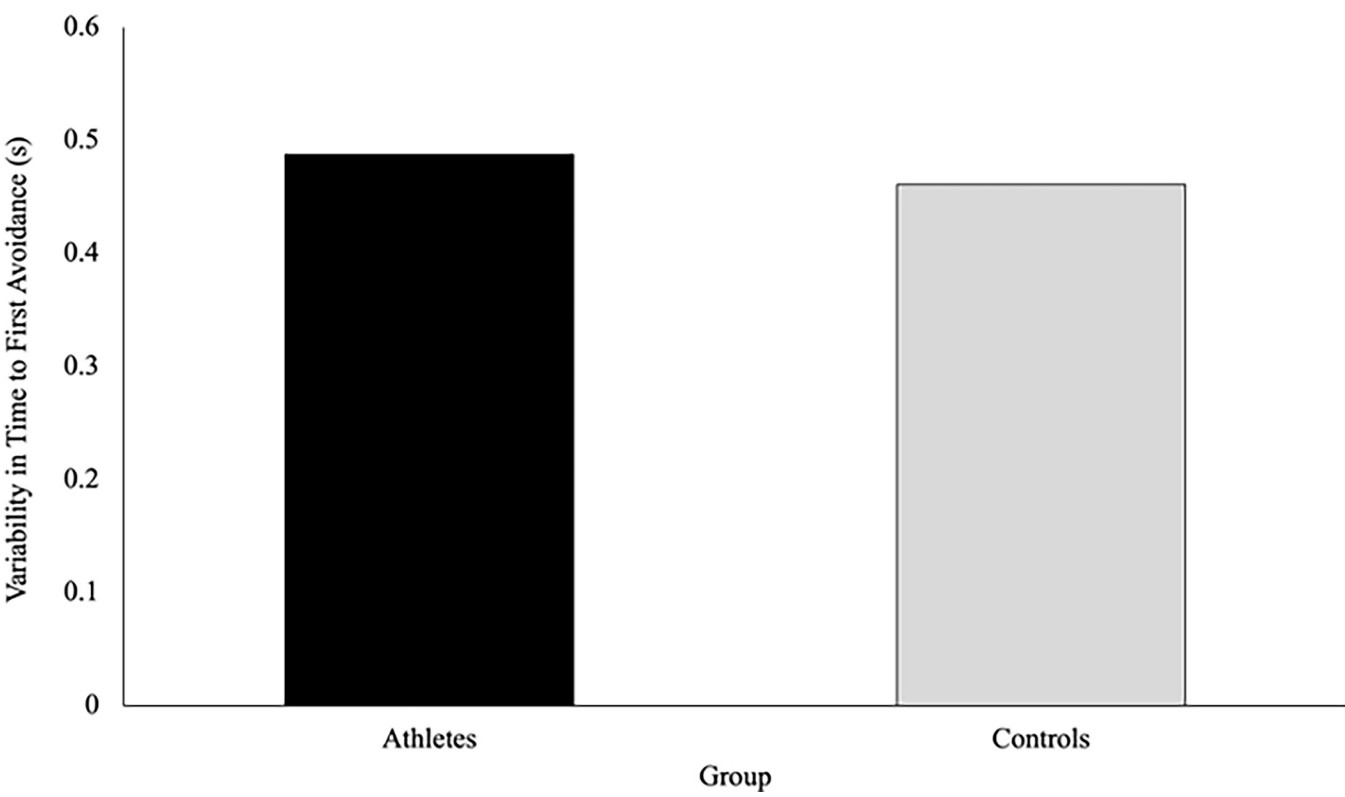

**Fig 6. Time to first avoidance (seconds) is a measure of the elapsed time between when the participants passed the trigger (1m) to when they initiated a behaviour change to avoid the virtual player.** This figure shows that there was no difference in the variability of time to first avoidance between athletes and controls (p = .613).

However, no significant interactions or main effects of direction or speed were observed for the variability of minimum clearance.

## Percentage of correct responses

There was a significant difference in the percentage of correct responses on the secondary task between athletes and controls, t(27.17) = -3.33, p = .003, d = -1.004. More specifically, the athlete group had a higher percentage of correct responses on the secondary task (95.1 ± 3.09%) compared to the controls (88.9 ± 7.86%) (**Fig 11**).

## Discussion

The current study sought to build upon previous research by examining whether sport-specific training influences individuals' action strategies while avoiding a virtual player approaching along a 45˚ angle and completing an attentionally demanding secondary task. It was expected that athletes would consistently avoid collisions later (i.e., get closer to the VP) while making fewer errors on the secondary task compared to the controls. The findings from this study revealed that athletes and controls employed similar action strategies, with athletes being more variable. However, the athletes performed better on the secondary task compared to the controls.

Contrary to our hypothesis, it was found that sport-specific training did not influence one's timing of avoidance as both groups initiated an avoidance behaviour at the same time (Fig 4). The lack of difference in the timing of avoidance behaviours between athletes and controls

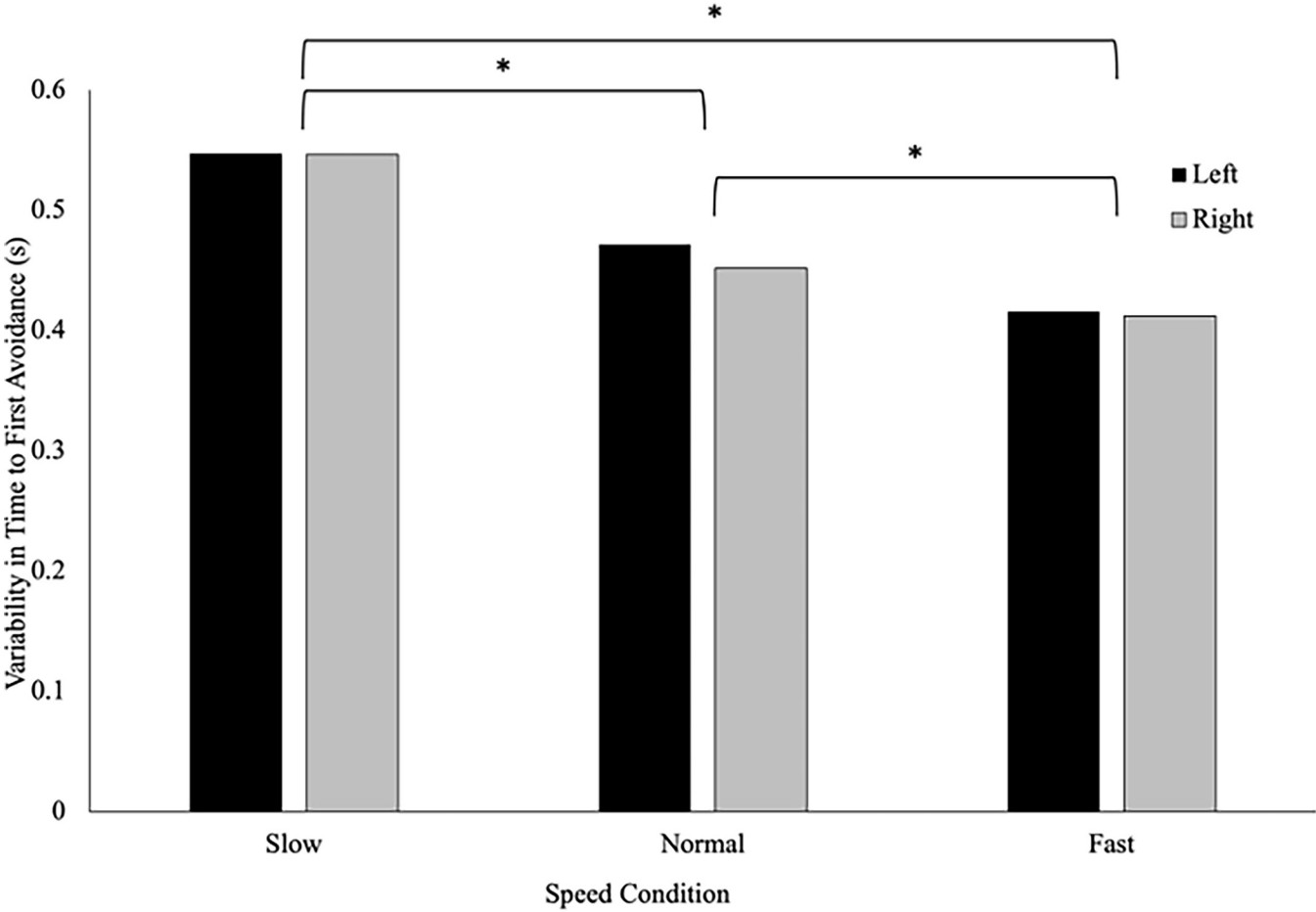

**Fig 7. The speed of the virtual player's approach influenced the variability of time to first avoidance, such that the variability in time to first avoidance behaviour decreased from the slow, to the normal, to the fast condition.** There was no effect of the direction of the virtual player's approach on variability in time of first avoidance.

could be due to the predictability of the VP's path. Pfaff and Cinelli [6] found that although both athletes and non-athletes relied on visual information (i.e., TTC and optical expansion threshold) to time their avoidance, athletes maintained a smaller and more consistent TTC when avoiding a pedestrian approaching along an unpredictable path (i.e., the participants did not know if the pedestrian was approaching from the left, right, or straight ahead until they had walked 2.5m). Thus, under unpredictable conditions, the avoidance behaviours of athletes seem to differ from their non-athlete counterparts, as training is thought to increase one's optical expansion threshold, allowing athletes to get closer to the potential collision (smaller TTC) prior to avoiding [6]. Conversely, when avoiding an obstacle approaching on a predictable path, individuals may not rely on an optical expansion threshold to time their avoidance, as an early deviation will always lead to a successful avoidance [5]. As such, it is possible that differences did not exist between athletes and controls in the current study, as participants were not relying on an optical expansion threshold to determine when to initiate an avoidance behaviour.

Additionally, the observed similarities in behaviour between athletes and controls suggests that the present study may not have provided a context sport specific enough to tease out the effects of training. Past research has demonstrated that specifically trained athletes may possess

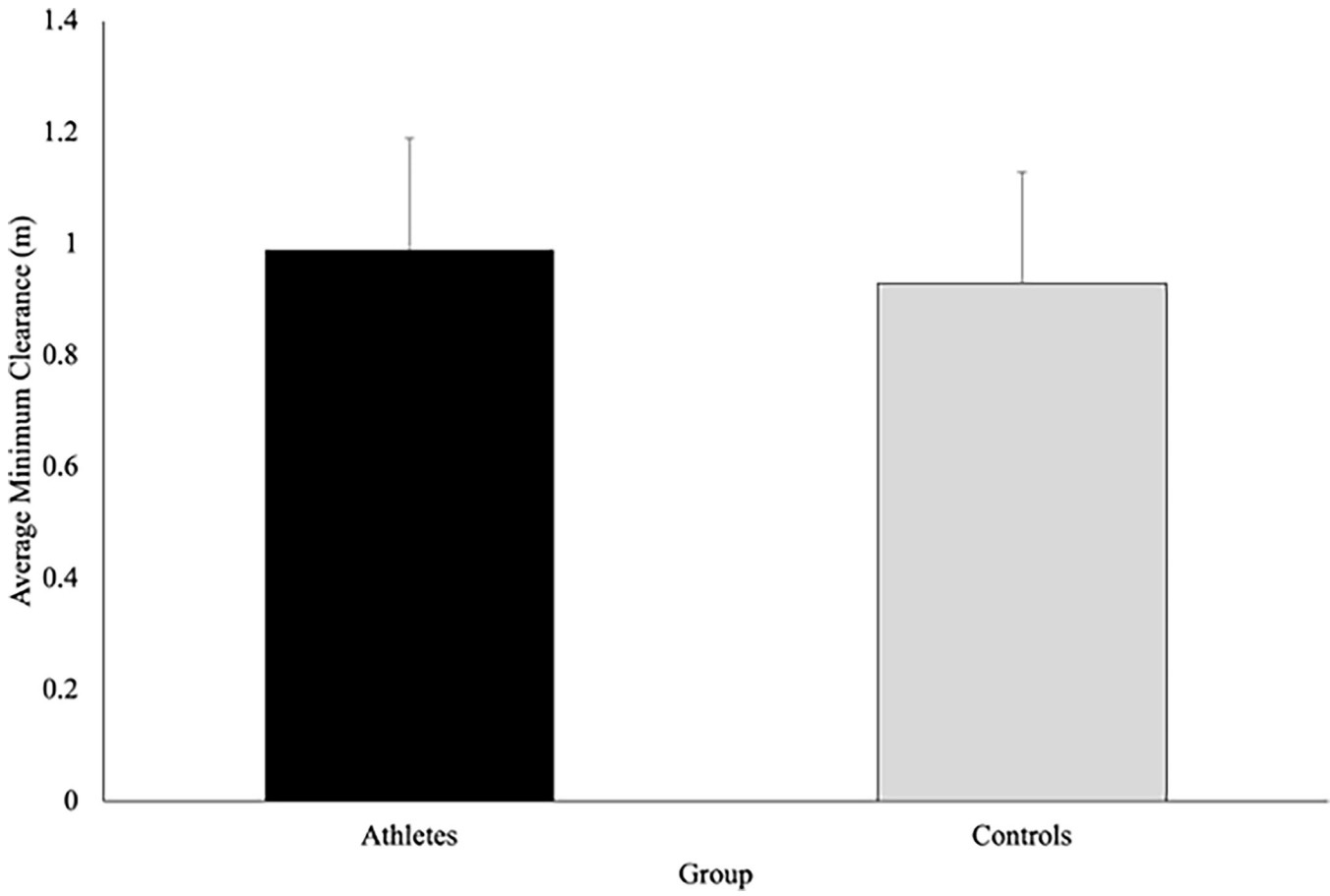

**Fig 8. Minimum clearance (meters; with SD bars) describes the minimum distance maintained between the participant and the virtual player.** This figure shows that the average minimum clearance was not significantly different between athletes and controls (p = .311).

enhanced perceptual, cognitive, and/or motor capabilities as a result of their training. However, further evidence has suggested that these neural adaptations are highly task-dependent and may reflect the demands of their respective activity [18,19]. For instance, Higuchi and colleagues [17] found that American football players had a better understanding of their body size and action capabilities (i.e., later onset and smaller magnitude of shoulder rotations) compared to untrained young adults while running through confined apertures (sport specific) but not while walking through apertures (non-sport-specific). Likewise, Pfaff and Cinelli [6] found differences in the avoidance behaviours of specifically trained rugby players and their untrained counterparts while avoiding a pedestrian on a constrained path and under unpredictable conditions. Similarly to Higuchi and colleagues [17], Pfaff & Cinelli [6] found that specifically-trained rugby players had a better understanding of their action capabilities as they both consistently and accurately avoided collisions later compared to their non-athlete counterparts. Collectively, the findings from these studies suggest that athletes do exhibit improved visuomotor skills which allow them to avoid obstacles more effectively than their non-athlete counterparts. However, these improved capabilities are highly task-dependent and may only be reflected when athletes are pushed to the boundaries of their performance by spatially and temporally confining them in ways in which they experience in their sport. Since the current task included a VP approaching along a predictable path, unconstrained movement, and had

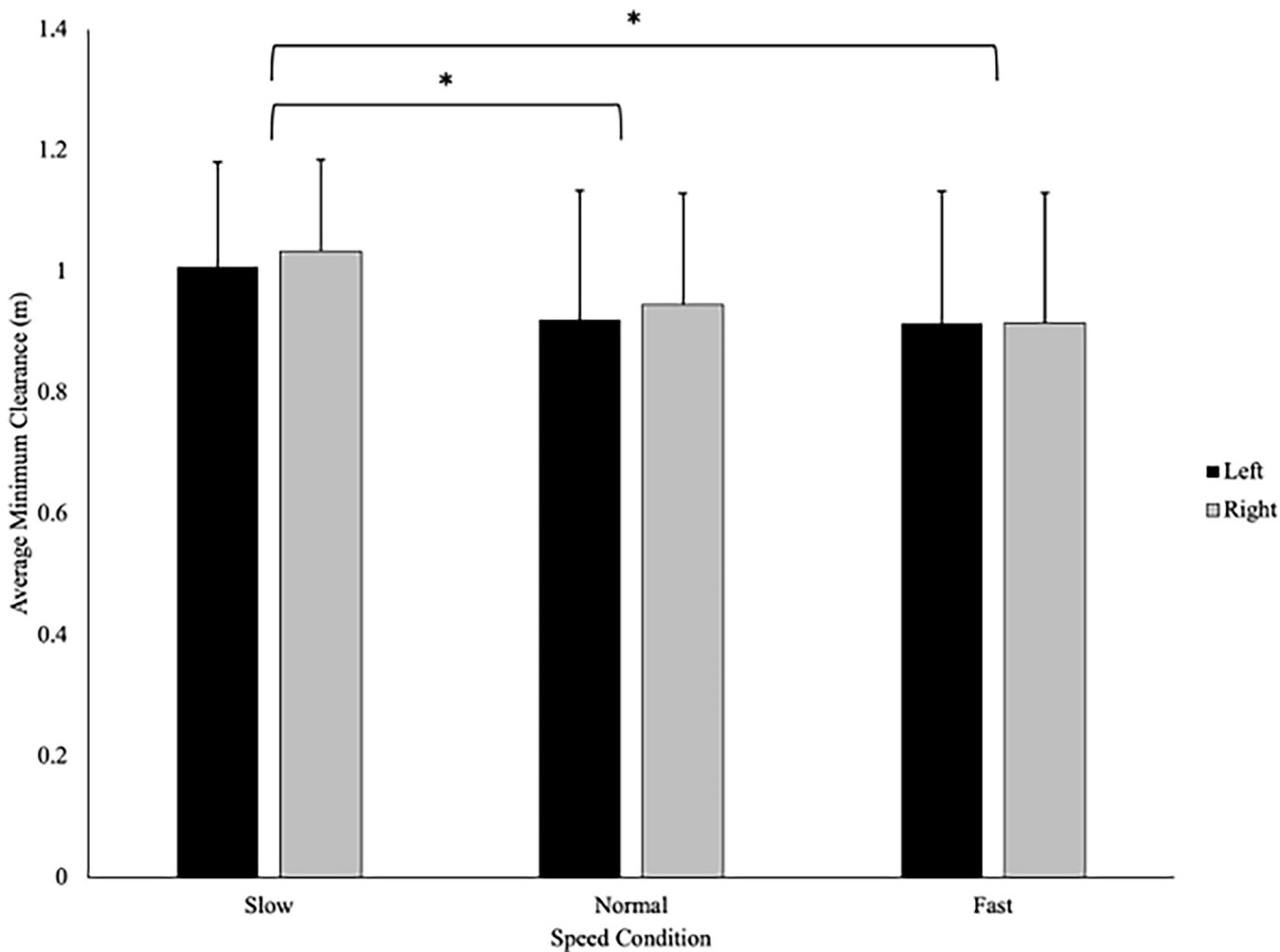

**Fig 9. The speed of the approaching virtual player influenced average minimum clearance as participants left more space in the slow condition, compared to the normal (p < .001) or fast (p < .001) speed conditions.** There was no effect of the direction of the virtual player's approach on average minimum clearance.

the participants walking, it is likely that the task was not sport-specific enough to tease out the differences between specifically trained athletes and controls.

Young adults (regardless of training) in the current study tended to initiate an avoidance behaviour at the same time irrespective of the speed of the approaching VP (Fig 5), suggesting that participants did not rely on optical expansion threshold to control *when* to initiate an avoidance behaviour. Similarly, Cinelli and Patla [5] found that young adults tended to initiate an avoidance behaviour at consistent point in space, while avoiding a human-like doll approaching at various speeds along a predictable path (180° collision course). In both studies, it appears that participants realized they could safely initiate an avoidance behaviour once they determined that the unresponsive object had begun moving towards them (i.e., optical expansion of object on retina) as opposed to waiting until the object reached an optical expansion threshold (due to the predictability of the obstacle's path). In contrast, Pfaff and Cinelli [7] found that participants initiated an avoidance behaviour at a smaller and more consistent TTC, while avoiding a head-on collision with a pedestrian approaching along an unpredictable path (i.e., pedestrian would deviate to participant's left, right, or continue straight). The

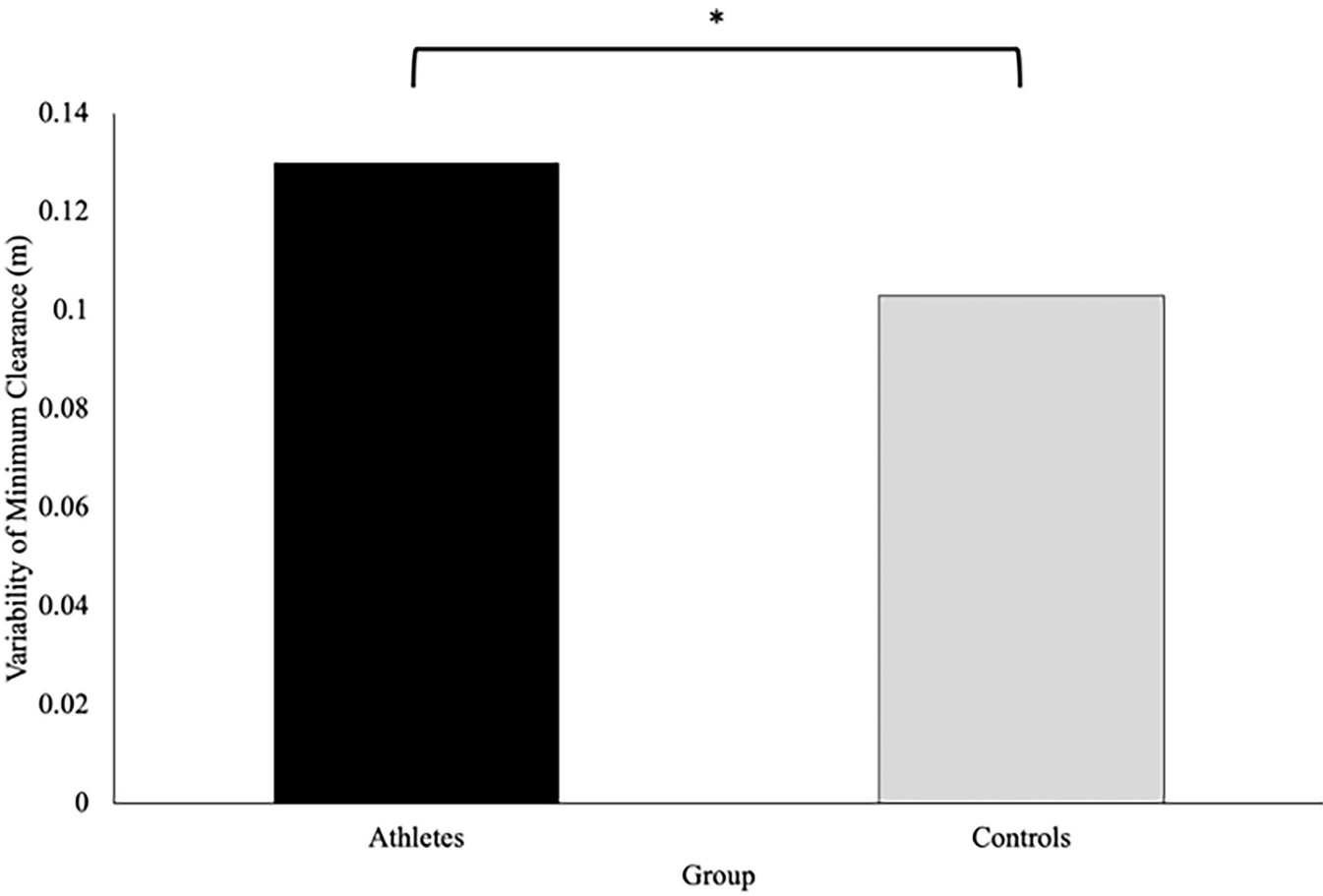

**Fig 10. Minimum clearance (meters) describes the minimum distance the participants maintained between them and the virtual player.** This figure shows that the variability of the minimum distance maintained by the participants was significantly different across groups, with athletes demonstrating greater variability compared to controls (p = .030).

findings from Pfaff and Cinelli [7] suggest that participants relied on an optical expansion threshold to time their avoidance, as an early deviation under unpredictable conditions could result in a collision. Thus, under unpredictable conditions, timing of avoidance may reflect one's optical expansion threshold, and the amount of time they need to safely avoid a collision with a person or object. Conversely, when the path of an approaching object is predictable (such as in the current study), participants (regardless of training) may solely rely on visual information to determine when an object has begun approaching, but initiate a behaviour change early, as this will always lead to a successful avoidance.

The findings from the current study also demonstrated that the average minimum clearance was similar between groups (Fig 8) which may suggest that sport-specific training does not influence one's personal space. This finding aligns with previous work by Pfaff and Cinelli [6] which demonstrated that although sport-specific training influenced the timing of avoidance (i.e., athletes avoided later), both rugby players and non-athletes exhibited similar spatial requirements (i.e., personal space) when avoiding a pedestrian on an unpredictable path. Similarly, past work has demonstrated that personal space was consistent among athletes and non-athletes when walking and running through apertures [20,21]. Collectively, from the current findings and those from previous work, it appears that sport-specific training may have minimal impact on an individual's personal space. As such, young adults (regardless of training)

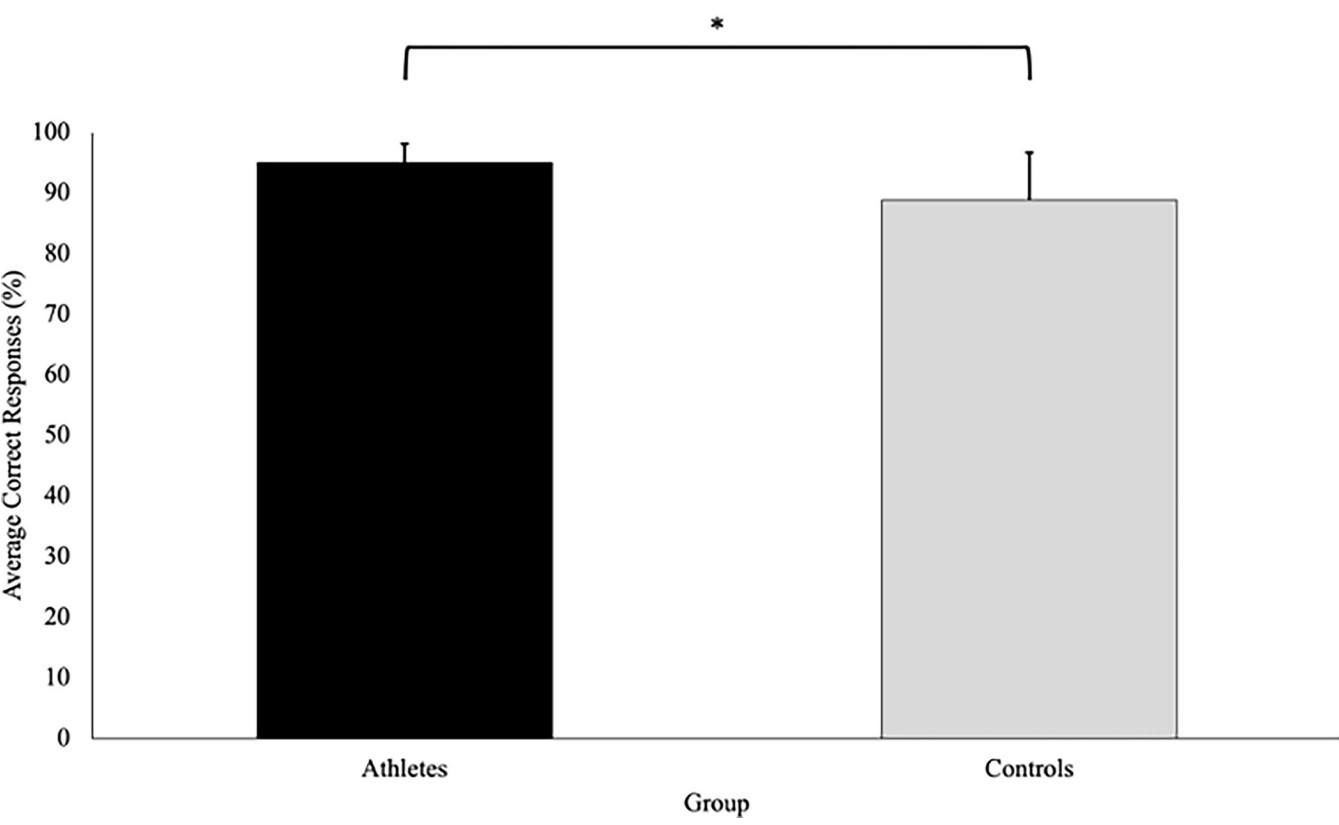

**Fig 11. Average correct responses (percentage; with SD bars) is a measure of the correct responses on the secondary task shown as a percentage of the total experimental trials completed (60 trials).** This figure shows that the athletes had a significantly higher percentage of correct responses on the secondary task compared to the controls (p = .003).

may be controlling their avoidance behaviour by maintaining a consistent minimum clearance distance (0.96m) between themselves and the VP during the normal and fast approach speeds, which is consistent with that found in previous work. For instance, it has been found that young adults control personal space, by maintaining a circular radius of approximately 1m between them and an opposing pedestrian during a stop-distance task [22]. Moreover, Orschiedt [23] demonstrated that participants maintained a minimum clearance distance of approximately 0.85m while performing a multi-task (walking, texting, and avoiding colliding with a moving pedestrian). Overall, it seems that young adults, regardless of their athletic training, maintain a consistent minimum clearance distance between themselves and an opposing person or object to maintain a consistent safety margin, even under complex multi-task conditions. This consistent safety margin may act as a protective zone to allow individuals adequate time to identify, evaluate, and respond to potential hazards within the environment.

Both groups in the current study maintained consistent minimum clearance distances for the normal and fast approach speeds (Fig 9) but increased the space when the VP approached at the slow speed, which may be due to social factors. When two individuals avoid colliding, they perceive the affordances of the opposing person [1]. As such, it is possible that the participants in our study may have attributed the slow walking speed of the VP (0.8x the participants' walking speed) to a population that often walks at similar speeds (i.e., an older adult), resulting in altered behaviours. Rapos and colleagues [24] demonstrated that young adults contributed more to the interaction when engaging with an older adult which resulted in increased clearance distances compared to when interacting with another young adult while interacting on a

90˚ collision course. The authors attributed these differences in behaviours to social norms and young adults' ability to perceive the affordances of older adults [24]. Therefore, person-specific characteristics of an approaching pedestrian such as age or movement profiles (slower walking speeds) may directly impact the space maintained by young adults, regardless of their training background.

Contrary to the original hypothesis, athletes were more variable in their action strategies compared to the controls (Fig 10). This finding may be supported by the theory of optimal feedback control which states that variance is only reduced in variables that are relevant to the outcome of the task [25]. As such, variance in human behaviour can exist without influencing the task outcome [25]. Further research has suggested that movement variability may be related to better performance on secondary tasks as well as improved adaptability to unexpected perturbations or obstructions within the environment [26]. Since the athletes in the current study successfully reached their goal without colliding with the virtual player, the current results may suggest that the increased variability in the athlete group may be related to an improved ability to adapt to the complexities of the environment, similar to what they would experience in a competition.

Even though our findings suggest that both groups had similar collision avoidance behaviours, the current study revealed that the specifically trained athletes performed significantly better (i.e., higher percentage of correct responses) on the secondary task compared to the controls (Fig 11). Our finding of improved secondary task performance suggests that specifically trained athletes may be better at performing attentional tasks in dynamic multitask environments. Individuals with greater athletic achieved higher scores on a battery of neurocognitive tests which measured visual attention and working memory (capacity and control) [27]. Given the close relationship between working memory capacity and attentional control, individuals with a high working memory capacity (i.e., specifically trained athletes) have a greater ability attend to concurrent tasks [28]. In the current study, individuals completed a highly complex collision avoidance task, while simultaneously identifying whether a shape changed above the VP's head, which may have been cognitively demanding. Consequently, individuals with a lower working memory capacity (i.e., untrained individuals) may have been forced to prioritize the task that possessed the greater risk of injury. Task prioritization has been demonstrated by the posture-first hypothesis which suggests that under dual-task conditions, individuals (primarily older adults) tend to prioritize balance or walking at the expense of a reduced secondary task performance [29]. Prioritizing balance over a cognitive task is likely a consequence of reduced attention allocation capabilities which limits one's ability to perform simultaneous tasks. Thus, it is possible that due to the complexity of the task in the current study, the controls may have adopted a similar behaviour to older adults to reduce cognitive load. Alternatively, since athletes possess an increased working memory capacity (and associated attentional control), it is possible that they are better at performing attention tasks in multitask environments. However, the authors note that although athletes demonstrated greater secondary task performance in the current study, behavioural differences did not emerge between the groups, which is likely due to the predictability and lack of sport-specificity of the task.

There are some limitations to the current study. First, the environment in which the study was conducted in may not have been created to differentiate athletes and untrained individuals. Past research has suggested that athletes must be tested under time or space constraints similar to their sport environment in order to examine behavioural differences [5]. Additionally, since the current study did not include a control block of trials involving only single-task performance, we are unable to examine cognitive performance measurements. Future studies should consider examining cognitive performance between athletes and untrained individuals during sport specific tasks (with space or time constraints) similar to the current study.

## Conclusions

Regardless of sport-specific training, individuals exhibit similar action strategies when avoiding an approaching virtual player at a 45˚ angle. However, athletes were more variable in their behaviours and performed significantly better on the secondary task, which suggests that they may be more adaptive with their actions and may perform better on attention tasks in dynamic environments. However, differences in avoidance behaviours did not emerge in this study, likely due to the predictability of the task and/or the lack of sport-specificity.

## Author Contributions

**Conceptualization:** Michael E. Cinelli.

**Formal analysis:** Michael E. Cinelli.

**Funding acquisition:** Michael E. Cinelli.

**Investigation:** Brooke J. Thompson, Michael E. Cinelli.

**Methodology:** Brooke J. Thompson, Michael E. Cinelli.

**Supervision:** Michael E. Cinelli.

**Writing – original draft:** Brooke J. Thompson, Michael E. Cinelli.

**Writing – review & editing:** Brooke J. Thompson, Michael E. Cinelli.

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
