## [Decision Letter · Decision Letter 0]

5 Jun 2024

PONE-D-23-31042The effects of sport-specific training on individuals action strategies while avoiding a virtual player approaching on a 45º angle while completing a secondary task.PLOS ONE

Dear Dr. Cinelli,

Thank you for submitting your manuscript to PLOS ONE. After careful consideration, we feel that it has merit but does not fully meet PLOS ONE’s publication criteria as it currently stands. Therefore, we invite you to submit a revised version of the manuscript that addresses the points raised during the review process.

We look forward to receiving your revised manuscript.

Kind regards,

Roxana Ramona Onofrei, PhD, MD

Academic Editor

PLOS ONE

 [Michael Cinelli received funding from the Natural Science and Engineering Research Council of Canada (NSERC) for this study].  

5. We note that you have indicated that there are restrictions to data sharing for this study. PLOS only allows data to be available upon request if there are legal or ethical restrictions on sharing data publicly. For more information on unacceptable data access restrictions, please see http://journals.plos.org/plosone/s/data-availability#loc-unacceptable-data-access-restrictions. 

Reviewers' comments:

Reviewer's Responses to Questions

**Comments to the Author**

1. Is the manuscript technically sound, and do the data support the conclusions?

Reviewer #1: Yes

Reviewer #2: Yes

2. Has the statistical analysis been performed appropriately and rigorously? 

Reviewer #1: Yes

Reviewer #2: Yes

3. Have the authors made all data underlying the findings in their manuscript fully available?

Reviewer #1: Yes

Reviewer #2: Yes

4. Is the manuscript presented in an intelligible fashion and written in standard English?

Reviewer #1: Yes

Reviewer #2: Yes

5. Review Comments to the Author

Reviewer #1: This study aims at comparing avoidance capabilities in athletes and control young subjects, more precisely in a 45° approaching path. The protocol is well described and data analysis is clear. The results were not all in accordance with the hypotheses and the authors are very honest in their discussion, not trying to find some micro parameter “coming from nowhere” to finally prove these hypotheses. The article would benefit from few minor corrections, answering the following questions.

Line 46 Avoiding collisions with humans is important within several sport settings but not in any sport (e.g. swimming, tennis, etc.). Please rephrase to avoid generality.

Line 63 What do you mean exactly by consistent size? Constant? It is surprising that it does depend on speed and maybe acceleration also.

Lines 79-84 You assess that avoiding later is a proof of larger capabilities in athletes compared to untrained people. Can you explain why later is better? In the following sentence you present it more as a strategy (some kind of dummy move). It does not actually prove that untrained people are not able to avoid later, but maybe only that they choose to avoid earlier.

Line 93 Could you explain a little bit more why a 45° approach is representative of sport-specific avoidance tasks?

Line 145 and foll. You mention that the shape change secondary task appears in half the trials. What was the condition in the other half?

Line 152 Were they explicitly asked to not prioritize one task over the other (verbally) or no instruction was given about prioritization?

Lines 206-208 The sentence sounds a bit strange as if some words were missing.

Lines 207-208 Some trials are missing: 1 in athletes, 6 in controls. Can you briefly explain why?

Line 282 Maybe your athlete group was not that different from the control one. What was their playing level, their training intensity (hours per week, years of practice, etc.)?

Reviewer #2: The authors evaluated whether a sport-specific training may have an influence on the individuals' collision avoidance behaviours during a sport-specific task conducted in a virtual reality setting.

The study covers an interesting topic, the rationale is well established, and the procedures are described in detail. However, although I believe that present study has merit there are some points that should be addressed. I hope that my comments will be useful to improve the overall quality of the manuscript.

Introduction

Line 87 – Please consider to expand the paragraph regarding the principle of "specificity” (a general task may lead to positive adaptations also in different outcomes) to reinforce the Authors’ hypotesis.

Here below some references to consider:

- Formenti D, Rossi A, Bongiovanni T, Campa F, Cavaggioni L, Alberti G, Longo S, Trecroci A. Effects of Non-Sport-Specific Versus Sport-Specific Training on Physical Performance and Perceptual Response in Young Football Players. Int J Environ Res Public Health. 2021 Feb 18;18(4):1962.

- Pagan JI, Bradshaw BA, Bejte B, Hart JN, Perez V, Knowles KS, Beausejour JP, Luzadder M, Menger R, Osorio C, Harmon KK, Hanney WJ, Wilson AT, Stout JR, Stock MS. Task-specific resistance training adaptations in older adults: comparing traditional and functional exercise interventions. Front Aging. 2024 Apr 30;5:1335534.

Line 100: Please consider to provide in Table 1 the number of years/months in terms of sport-specific background within the trained athletes group.

Line 115: Did the Authors performed also cognitive performance measurements (i.e., executive control, perceptual speed)?

Line 413: Please consider to add a specific paragraph regarding the main limitations as well as the future perspectives derived from the Author’s’ results.

6. PLOS authors have the option to publish the peer review history of their article (what does this mean?). If published, this will include your full peer review and any attached files.

Reviewer #1: No

Reviewer #2: No

---

## [Author Response · Author response to Decision Letter 0]

3 Jul 2024

** The line numbers listed in this document refer to the line numbers in the “Manuscript” file. The line numbers in the “Revised Manuscript With Track Changes” file have been altered due to the track changes do not match the ones in the “Manuscript File” **

Reviewer 1 

Line 46 Avoiding collisions with humans is important within several sport settings but not in any sport (e.g. swimming, tennis, etc.). Please rephrase to avoid generality.

Thank you to the reviewer for the comment regarding the general statement regarding sports. We agree with the reviewer that collision avoidance is not important in all sports. Thus, we have changed the sentence on Line 46 to say “in many sport settings” to suggest that not all sports involve collision avoidance (Line 44). 

Line 63 What do you mean exactly by consistent size? Constant? It is surprising that it does depend on speed and maybe acceleration also.

Thank you to the reviewer for the comment regarding the size of one’s personal space. While navigating an avoidance, multiple studies have found consistency in the amount of space individuals tend to leave between themselves and another object or person to maintain a safety zone. Thus, we believe the word consistent describes the safety zone individuals tend to leave between themselves and an opposing person/object. We included a clarifying statement in the manuscript to outline that this has been shown in previous work (Line 61). 

Lines 79-84 You assess that avoiding later is a proof of larger capabilities in athletes compared to untrained people. Can you explain why later is better? In the following sentence you present it more as a strategy (some kind of dummy move). It does not actually prove that untrained people are not able to avoid later, but maybe only that they choose to avoid earlier.

Thank you to the reviewer for the comment regarding the adapted collision avoidance behaviours in specifically trained athletes. We agree that although athletes may adapt their avoidance behaviours as a strategy to successfully act within their sport, this is not necessary an improved avoidance compared to untrained individuals. Therefore, we have changed the language to describe this behaviour as a sport-specific strategy used to extract important information from the environment (Line 72-73). 

Line 93 Could you explain a little bit more why a 45° approach is representative of sport-specific avoidance tasks?

Thank you to the reviewer for their question regarding the 45° approach. In a sport setting, collisions do not always occur on 180° or 90° angle which is the angle of approach that is traditionally examined in the collision avoidance literature. Thus, we wanted to include an approach angle that was less traditional, more similar to a sport situation and one that was difficult to avoid simulating what they would experience in sports. 

Line 145 and foll. You mention that the shape change secondary task appears in half the trials. What was the condition in the other half?

Thank you to the reviewer for their question about the secondary task. On 50% of the trials, the shape above the VP’s head would change to one of 4 shape options. On the remaining 50% of trials, there would be no shape change (the shape would stay as a yellow square the whole time). However, we did not examine the effect of the secondary task, as the trials were randomized, and thus, participants were always looking for the shape change regardless of whether it changed or not. To clarify this, we added a sentence stating that the shape would remain unchanged (as a yellow square) on half of the trials (Line 146-147).

Line 152 Were they explicitly asked to not prioritize one task over the other (verbally) or no instruction was given about prioritization?

Thank you to the reviewer for their question about the instructions given to the participants. The participants were not asked to prioritize either task. They were asked to “walk to the goal while avoiding colliding with the VP and to report whether a shape changed above either of the VPs’ heads.” (Line 147-149). Line 151-152 in the manuscript describes that participants were not asked to prioritize either task. 

Lines 206-208 The sentence sounds a bit strange as if some words were missing.

Thank you to the reviewer for pointing out the errors within the sentence. The sentence has been edited to provide more clarity (Line 203-206). 

Lines 207-208 Some trials are missing: 1 in athletes, 6 in controls. Can you briefly explain why?

Thank you to the reviewer for their question regarding the missing trials. There was a tracker error on the HMD which resulted in 7 trials being omitted from the analysis. Mean substitution was performed to correct obscure data points resulting from a tracking error. To complete mean substitution, the average of two unobstructed trials in the same condition (the ones above and below) were substituted for the obstructed trial. The substituted value was used for calculating each of the dependent variables. For the participants with obstructed trials, the percentage of correct responses was calculated using the total number of unobstructed trials. We have changed the reported values to percentages in the paper to avoid confusion (Line 205-206). 

Line 282 Maybe your athlete group was not that different from the control one. What was their playing level, their training intensity (hours per week, years of practice, etc.)?

Thank you to the reviewer for their comment about the level of training in the athlete group. The athletes in the current study were varsity athletes on the roster for a team and currently playing. We did not collect data on the years of experience, however, past research in our lab has not found a correlation between years in sport and behavioural measures. 

Reviewer 2

Line 87 – Please consider to expand the paragraph regarding the principle of "specificity” (a general task may lead to positive adaptations also in different outcomes) to reinforce the Authors’ hypothesis.

Here below some references to consider:

- Formenti D, Rossi A, Bongiovanni T, Campa F, Cavaggioni L, Alberti G, Longo S, Trecroci A. Effects of Non-Sport-Specific Versus Sport-Specific Training on Physical Performance and Perceptual Response in Young Football Players. Int J Environ Res Public Health. 2021 Feb 18;18(4):1962.

- Pagan JI, Bradshaw BA, Bejte B, Hart JN, Perez V, Knowles KS, Beausejour JP, Luzadder M, Menger R, Osorio C, Harmon KK, Hanney WJ, Wilson AT, Stout JR, Stock MS. Task-specific resistance training adaptations in older adults: comparing traditional and functional exercise interventions. Front Aging. 2024 Apr 30;5:1335534.

Thank you to the reviewer for their comments regarding the specificity of the task. We have expanded the introduction to discuss how athletes tend to have an improved performance when the task is performed in an environment closely related to the environment in which they train (Line 88-90). 

Line 100: Please consider to provide in Table 1 the number of years/months in terms of sport-specific background within the trained athletes group.

Thank you to the reviewer for their comment regarding the number of years/months of sport specific background within the trained athletes group. The athletes in the current study were varsity athletes on a team roster and currently playing and were therefore, highly trained in their sport. We did not collect data on the number of years of sport specific background, however, past research in our lab has not found a correlation between years in sport and behavioural measures. 

Line 115: Did the Authors performed also cognitive performance measurements (i.e., executive control, perceptual speed)?

Thank you to the reviewer for their question about the cognitive performance measurements. In the current study, we did not have a control trial where there was no secondary task. Therefore, we were unable to measure cognitive performance. This has been listed as a direction/perspective for future research building from this study (Line 415-417). 

Line 413: Please consider to add a specific paragraph regarding the main limitations as well as the future perspectives derived from the Authors’ results.

Thank you to the reviewer for the comment regarding limitations and future directions. A paragraph has been added to the end of the discussion (Line 412-419).

---

## [Decision Letter · Decision Letter 1]

11 Sep 2024

The effects of sport-specific training on individuals action strategies while avoiding a virtual player approaching on a 45º angle while completing a secondary task.

PONE-D-23-31042R1

Dear Dr. Cinelli,

We’re pleased to inform you that your manuscript has been judged scientifically suitable for publication and will be formally accepted for publication once it meets all outstanding technical requirements.

Kind regards,

Anwar P.P. Abdul Majeed

Academic Editor

PLOS ONE

Additional Editor Comments (optional):

Reviewers' comments:

Reviewer's Responses to Questions

**Comments to the Author**

1. If the authors have adequately addressed your comments raised in a previous round of review and you feel that this manuscript is now acceptable for publication, you may indicate that here to bypass the “Comments to the Author” section, enter your conflict of interest statement in the “Confidential to Editor” section, and submit your "Accept" recommendation.

Reviewer #1: All comments have been addressed

Reviewer #2: All comments have been addressed

2. Is the manuscript technically sound, and do the data support the conclusions?

Reviewer #1: (No Response)

Reviewer #2: Yes

3. Has the statistical analysis been performed appropriately and rigorously? 

Reviewer #1: (No Response)

Reviewer #2: Yes

4. Have the authors made all data underlying the findings in their manuscript fully available?

Reviewer #1: (No Response)

Reviewer #2: Yes

5. Is the manuscript presented in an intelligible fashion and written in standard English?

Reviewer #1: (No Response)

Reviewer #2: Yes

6. Review Comments to the Author

Reviewer #1: The authors provided clear and convincing answers to any of my questions. The article is now fully suitable for publication

Reviewer #2: The Authors addressed all the comments properly. A last suggestion: please consider to add proper citations within the introduction section when dealing with training "specificity”.

Here below some references to consider:

- Formenti D, Rossi A, Bongiovanni T, Campa F, Cavaggioni L, Alberti G, Longo S, Trecroci A. Effects of Non-Sport-Specific Versus Sport-Specific Training on Physical Performance and Perceptual Response in Young Football Players. Int J Environ Res Public Health. 2021 Feb 18;18(4):1962.

- Pagan JI, Bradshaw BA, Bejte B, Hart JN, Perez V, Knowles KS, Beausejour JP, Luzadder M, Menger R, Osorio C, Harmon KK, Hanney WJ, Wilson AT, Stout JR, Stock MS. Task-specific resistance training adaptations in older adults: comparing traditional and functional exercise interventions. Front Aging. 2024 Apr 30;5:1335534.

7. PLOS authors have the option to publish the peer review history of their article (what does this mean?). If published, this will include your full peer review and any attached files.

Reviewer #1: No

Reviewer #2: No

---

## [Editor Report · Acceptance letter]

16 Oct 2024

PONE-D-23-31042R1 

PLOS ONE

Dear Dr. Cinelli, 

I'm pleased to inform you that your manuscript has been deemed suitable for publication in PLOS ONE. Congratulations! Your manuscript is now being handed over to our production team.

Kind regards, 

on behalf of

Dr. Anwar P.P. Abdul Majeed 

Academic Editor

PLOS ONE